# Impact of artificial light at night on diurnal plant-pollinator interactions

Simone Giavi [1], Colin Fontaine[2] & Eva Knop [1,3] ✉

Artificial light at night has rapidly spread around the globe over the last decades. Evidence is increasing that it has adverse effects on the behavior, physiology, and survival of animals and plants with consequences for species interactions and ecosystem functioning. For example, artificial light at night disrupts plant-pollinator interactions at night and this can have consequences for the plant reproductive output. By experimentally illuminating natural plant-pollinator communities during the night using commercial street-lamps we tested whether light at night can also change interactions of a plant-pollinator community during daytime. Here we show that artificial light at night can alter diurnal plant-pollinator interactions, but the direction of the change depends on the plant species. We conclude that the effect of artificial light at night on plant-pollinator interactions is not limited to the night, but can also propagate to the daytime with so far unknown consequences for the pollinator community and the diurnal pollination function and services they provide.

[1] Departement of Agroecology and Environment, Agroscope, Zürich, Switzerland. [2] Centre d'Ecologie et des Sciences de la Conservation, CESCO, Muséum National d'Histoire Naturelle—CNRS—Sorbonne Université, Paris, France. [3] Departement of Evolutionary Biology and Environmental Sciences, University of Zürich, Zürich, Switzerland. ✉email: eva.knop@ieu.uzh.ch

A "luminous fog" of artificial light at night is enveloping the locations inhabited by humans, with about 18.7% of the world's terrestrial surfaces (excluding Antartica) currently being exposed to it[1]. Furthermore, the area experiencing direct emissions from artificial light sources is estimated to expand at about 2–6% per year while already illuminated areas become brighter at a similar rate[2]. In addition to effects on the physiology and behavior of organisms with consequences for mortality, reproduction, species abundance, and community composition[3–7], artificial light at night affects species interactions and ecosystem functioning[8]. To date, most research on the impact of artificial light at night on species interactions has focused on altered foraging strategies of night-active predators, in particular birds[9], bats[10–13], and spiders[14]. More recently also plant–herbivore interactions have been found to be altered by artificial light at night[15–17], which can even be most pronounced in dark areas adjacent to artificially illuminated areas[18]. Further, artificial light at night has been shown to disrupt nocturnal plant–pollinator interactions[19,20], with negative consequences for plant reproductive output[19]. Interestingly, also a positive effect of artificial light at night on the reproductive output has recently been reported[21], which suggests more complex indirect effects of artificial light at night, probably involving diurnal pollinator communities or other organisms such as herbivores or predators. Nonetheless, by merging day- and nighttime plant–pollinator interaction networks, Knop et al.[19] showed that diurnal and nocturnal pollinator communities are linked in a way that they should favor the spread of a nocturnal perturbation to the diurnal community. However, we still miss evidence that effects of artificial light at night can actually spread to diurnal interactions.

Here we therefore experimentally test whether the effect of artificial light also affects diurnal plant–pollinator interactions. In 2016, we sampled diurnal plant–pollinator interactions on 12 unmanaged meadows, of which 6 were experimentally illuminated at night using commercial LED street lamps (lit treatment sites) and 6 were left untreated (dark control sites) using a paired approach (see "Methods"). The paired approach was such that during daytime, we simultaneously quantified plant–pollinator interactions along 100-m transects on a pair of treatment and control sites (37 sampling events with each a control and treatment site sampled simultaneously; see "Methods"). This allows minimizing the influence of factors other than the light treatment, which potentially could influence the number of plant–pollinator interactions, such as weather conditions. In the analysis, we focus on the interactions between 21 naturally occurring plant species regularly present across our study sites, and on insect groups acknowledged to be pollinators, namely Diptera, Hymenoptera, and Coleoptera (2384 interactions in total; see "Methods"). Interactions involving other insect groups or plant species present on only a few sites or in very low abundances were also recorded but not included because their number was too low to be analysed (see "Methods").

We test whether artificial light at night changes the number of diurnal plant–pollinator interactions (response variable) using a model that includes the following explanatory variables: plant species (21 levels), light treatment (dark versus lit), insect group (Diptera, Hymenoptera, and Coleoptera), the interactions among all of them, and the abundance of plant species as co-variable. Our analysis shows that artificial light can alter the number of diurnal plant–pollinator interactions, but the direction of the change depends on the plant species involved.

## Results

Of the 2384 diurnal plant–pollinator interactions, 984 involved Hymenoptera, 1119 Diptera, and 281 Coleoptera. Plant abundance was positively related to the number of diurnal plant–pollinator interactions, and the number of plant–pollinator interactions differed between plant species and pollinator group (Table 1). We found a significant interaction between the light treatment and plant species indicating that depending on the plant species, artificial light at night significantly altered the number of plant–pollinator interactions during daytime (Table 1 and Fig. 1a). Three plant species received significantly (i.e., no overlap of the CI with zero) less interactions and one species received significantly more interactions under the light treatment (Fig. 1a). In addition, one species showed a trend for receiving less interactions, one species for more interactions, respectively (Fig. 1a). Interestingly, the plant-specific effect of artificial light at night on diurnal plant–pollinator interactions sometimes also varied depending on the insect group involved in the interaction as highlighted by the significant interaction treatment:plant:insect group (Table 1, Fig. 1b–d, and Supplementary Table 2). For example, the number of interactions of *Geranium sylvaticum* was similar between lit and dark meadows (Fig. 1a) but the composition of insect visitors differed, with significantly less visits from Diptera on lit meadows and a trend for more visits from Coleoptera (Fig. 1b, d). Such a pattern of a similar number of total interactions on lit and dark meadows, but insect-group-specific responses to the light treatment, was also found for other plant species, such as *Centaurea sp.* and *Angelica sylvestris*. This could potentially lead to a change of the quality of the pollination service the pollinators provide.

## Discussion

In 19% of the plant species investigated, artificial light at night altered the total number of pollinator visits received during daytime, but the direction of the effect was plant-specific. Thus, the effect of artificial light at night is not limited to plant–pollinator interactions at night, but can also spread to diurnal plant–pollinator interactions. Interestingly, more plant species of which the diurnal interaction frequencies were altered, showed a significant reduction in interactions, whereas only one plant species showed an increase due to artificial light at night. This suggests that the effect of artificial light at night generally leads to a reduction of plant–pollinator interactions during daytime with potential knock-on consequences on pollination, though more studies in more systems are needed to confirm this

**Table 1 Effect of artificial light at night on diurnal plant–pollinator interactions.**

|  | Sum of squares | d.f. | *F* value | *P* |
|---|---|---|---|---|
| Plant species | 80.846 | 20 | 15.08 | **<0.001** |
| Treatment | 0.270 | 1 | 1.01 | 0.316 |
| Pollinator group | 32.904 | 2 | 61.36 | **<0.001** |
| Plant abundance | 124.247 | 1 | 463.44 | **<0.001** |
| Plant species:treatment | 12.149 | 20 | 2.27 | **0.001** |
| Plant species: pollinator group | 113.564 | 40 | 10.59 | **<0.001** |
| Treatment: pollinator group | 0.334 | 2 | 0.62 | 0.536 |
| Plant species: treatment: pollinator group | 17.496 | 40 | 1.63 | **0.008** |

Anova table of the general linear mixed effects model testing for the effect of artificial light at night (control versus illuminated), plant abundance (scaled logarithm), plant species (21 levels), insect group (three levels: Diptera, Hymenoptera and Coleoptera), and the interactions between treatment and plant species on the number of plant–pollinator interactions (response variable, log-transformed). Estimate values are provided in Supplementary Data 1. *P* < 0.05 are presented in bold. Source data are provided as a Source Data file.

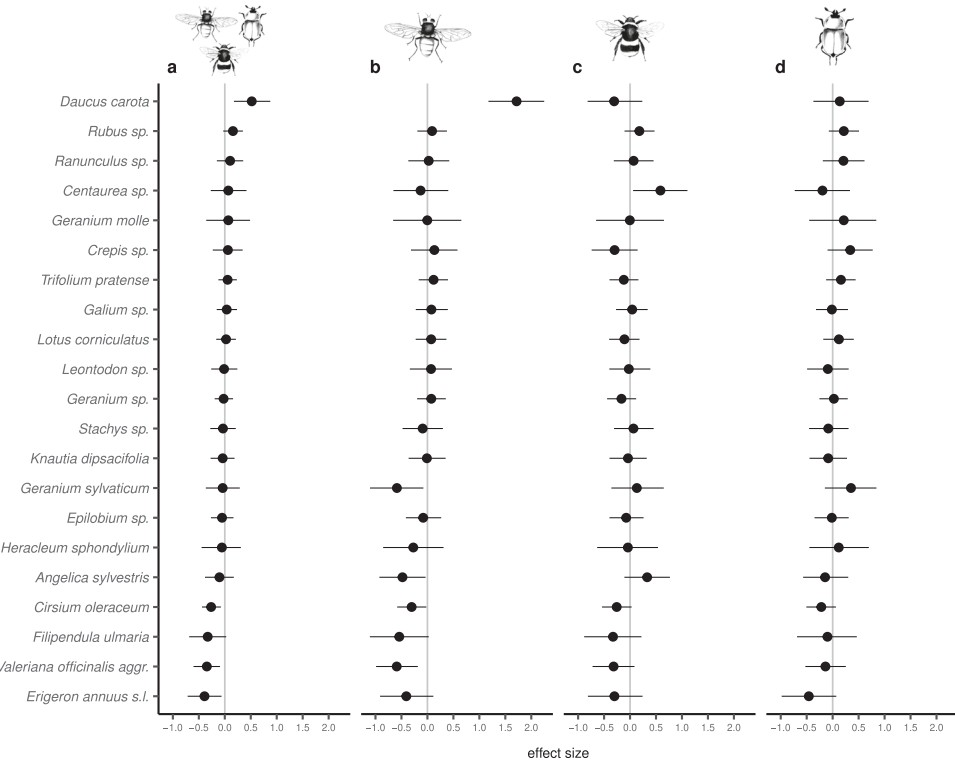

**Fig. 1 Effect of artificial light at night on diurnal plant–pollinator interactions.** Plant-specific estimated effects (median ± 95% credible interval, computed from the marginal posterior distribution of the model parameters) of the light treatment on the total number of plant–pollinator interactions (**a** all insect groups together), and separate for the different insect groups (**b** Diptera, **c** Hymenoptera, **d** Coleoptera). As the response variable was log-transformed to fulfill the model assumptions (Gaussian error distribution), the back transformed number of interactions at lit sites equals to the number of interactions at unlit sites × exp(effect size). For example, an average of 10 interactions at unlit sites with an effect size of −1 translates in 10 × exp(−1) interactions at lit sites (i.e., 3.68 interactions).

trend. Further than changes in the total number of interaction per plants species, our results indicate that insect orders responded differently to the light treatment, resulting in different composition of the visiting fauna for some plant species. Such changes, even if not translated into changes in the total number of visits received due to compensatory responses, might affect the plant reproductive output as the per visit pollination effectiveness might differ among insect orders.

Several indirect pathways might explain these findings. First, the effect of artificial light at night on diurnal plant–pollinator interactions might be due to a change of the expression of floral traits, which are key for mediating plant–pollinator interactions[22]. We are currently not aware of a study quantifying a change of the expression of floral traits due to artificial light at night at the intensities and spectral composition caused by public lightings, except for one study that found a reduction in flower density in *Lotus pedunculatus*[23]. However, the spectrum and intensity of LED street lamps is known to have the potential to affect the physiological processes linked to the seasonal and daily timing of plants, which is also key for the expression of floral traits[24]. Floral traits like scent emission, which often show very pronounced diel expression patterns, might be particularly affected by light[25]. Thus, our light treatment might have modified the expression of floral traits with consequences for plant–pollinator interactions during daytime. This idea is supported by previous studies highlighting the importance of circadian rhythms in plants[26,27] and pollinators[28] for successful interactions, and by reviews discussing the potential impact of artificial light at night on those rhythms and on the resulting pollination success[24,25]. For example, experiments with genetically modified *Nicotiana attenuata* (Solanaceae) plants in which

different circadian clock genes were silenced, showed, that the circadian clock affects interactions with diurnal and nocturnal pollinators and, depending on the available pollinator community, results in different fitness outcomes[27]. Finally, we found that the plant-specific effects of artificial light at night on plant–pollinator interactions varied between insect groups. This might be a consequence of the fact, that different pollinator groups respond differently to floral traits[22], which potentially are differently affected by artificial light at night.

Alternatively to a direct impact on plant physiology, our results might be due to more complex pathways involving other trophic levels such as herbivores. Herbivores are known to affect the attractiveness of plants to pollinators with consequences for flower visits and pollination success[29,30]. For example, herbivory has previously been found to reduce the number of pollination visits[31–34], though the opposite has also been found[35,36] as well as no effect at all[37,38]. Thus, potential herbivore-mediated effects on plant attractiveness to pollinators might be another reason for both positive and negative effects of artificial light at night on individual plant species. A previous study has found that light, depending on its wavelength and on the presence of predators, can decrease abundance of herbivores[15]. Further, artificial light at night can directly (by decreasing larvae weight) and indirectly (by increasing host plant leaf toughness) decrease herbivores performance[16]. On the other hand, also an increase in herbivory under artificial light at night has been found[17]. Thus, the reduced and increased number of diurnal plant–pollinator interactions might be due to changed herbivory at night.

Finally, another pathway for increased plant–pollinator visitations during daytime due to artificial light at night might involve nocturnal pollinators. Previous studies suggest that nocturnal

plant–pollinator interactions are disrupted due to artificial light at night[19,20], which might decrease the nectar depletion by nocturnal pollinators at night and thus in turn might increase diurnal plant–pollinator interactions.

While previous studies brought evidence that artificial light at night disrupts plant–pollinator interactions during the night[19,20] with potential consequences for the plant–pollinator community during daytime[19], here we bring evidence that such knock-on effects actually occur. In contrast to the few previous studies which focused on such a spreading effect of artificial light at night to the day, and which found either for single species a prolongation of their activity during daytime[39] or a shift in the abundance of invertebrate specimens within taxonomic groups[5], our results expand these findings to species interactions at the community level. Although the effect of artificial light at night varied among plant species and pollinator groups, the alteration of the interactions could have consequences for community stability and functioning.

Artificial light at night is increasingly recognized as a global change driver[40]. Our results show that further than directly affecting the functioning of nighttime ecological activities, the impact is amplified by secondary responses propagating through the system of interacting species thereby affecting other species of the community, which are not directly exposed to the global change driver. This is similar to what has been observed with other global change drivers[41] and stresses the need to further investigate the consequences at community level, including the full day and night periods. While the mechanisms underlying this effect of artificial light at night on diurnal plant–pollinator communities still need to be elucidated in future, another research avenue is to look for the interactive effects between artificial light at night and other recognized global change drivers[41].

We conclude that the consequences of artificial light at night are not limited to the night, but can also propagate to daytime by altering species interaction and potentially the ecosystem functions relying on it, such as pollination. This is alarming given the many global change stressors diurnal pollinators already experience.

## Methods

**Experimental design.** In 2016, twelve independent ruderal meadows were selected as sampling sites in the Prealps of Switzerland. This region has low levels of light emission with a radiance lower than $0.25 \times 10^{-9}$ W sr$^{-1}$ cm$^{-1}$ (data from http://www.lightpollutionmap.info). All sites were located at least 100 m away from minor light sources and 500 m away from major light sources, such as illuminated sport grounds. Average distances among sites were $11.94 \pm 0.83$ km, and we ensured that they were all unmanaged.

On six of the 12 sites we experimentally installed a commercial LED street lamp (Schréder GmbH, type: AMPERA MIDI 48 LED, color temperature: neutral white (4000 K), nominal LED flux: 6800 lm) on 6-m high poles. Hereafter we refer to the sites with an experimentally installed LED street lamp as "illuminated sites." The other sites were left dark ("dark sites" hereafter) but equipped with a fake street lamp to provide comparable conditions. Light intensity on illuminated sites followed a negative exponential curve as function of the distance from the lamp dropping from $75.73 \pm 1.54$ lux just under the lamp (<2 m) to $2.67 \pm 0.19$ lux $10 \pm 1.0$ m away (Supplementary Fig. 2). Transect for pollinator sampling was 100-m long with the street lamp in the middle, resulting in pollinator sampling all along the light intensity gradient. Light measurements were performed at least 2 h after sunset and in the absence of moon, using a universal photocurrent amplifier with computer interface (by Czibula & Grundmann GmbH, www.photo-meter.com), always keeping the sensor at 70 cm of height (the average flowers height) and pointing upward. Control and illuminated sites were paired according to spatial proximity and each site pair was sampled simultaneously to control for environmental variations (see Supplementary Table 1). We will refer to "sampling event" for the sampling of a site pair.

**Assessment of plant–pollinator interactions.** Plant–pollinator interactions were assessed on each site pair between 6 and 7 times, for a total of 37 sampling events. Sampling events occurred during the afternoon between 13:00 and 17:00 when a maximum of pollinators were active, except for five samplings that were performed in the morning. On average, the same site was sampled every 15 days. Each sampling event was subdivided into six sampling rounds that took place every 30 min. During each round, we collected all flower visitors actively touching the reproductive organs of a receptive flower within the area of 1 m on both sides of a 100-m transect (one transect per site) walking at a steady speed[42]. For the analysis, pollinators were defined as insects belonging to the groups of Diptera, Hymenoptera, and Coleoptera (excluding those belonging to the Carabidae family). Although also other groups of insects are known to be pollinators, they were excluded from the analysis as the number of interactions was too low to be analysed at the group level (43 Hemiptera, 14 Lepidoptera, 2 Dermaptera, 2 Mecoptera, and 1 Neuroptera). At each interaction, plant species was recorded, whereas pollinator was captured using a hand net, transferred in a vial and frozen.

**Assessment of plant abundance.** As plant species and abundance varied considerably among sites from one sampling event to another, the surface covered by the flowers of a given plant species along the transect (1 m on both sides of it) was estimated using a surface unit of a circle of about 5 cm of diameter shortly before each sampling event. For each plant species, we assigned a discrete number of units that were used as a proxy for floral cover. Most abundant and widespread plant was *Cirsium oleraceum* (Asteraceae), followed by other species being abundant but not present on all sampling sites: *Angelica sylvestris* (Apiaceae), *Eupatorium cannabinum* (Asteraceae), *Erigeron annuus s.l.* (Asteraceae), and *Filipendula ulmaria* (Rosaceae).

**Analysis.** Interaction data were filtered for 21 plant species ("selected plants" hereafter) that were regularly found flowering on the sites during the sampling events, namely during at least five samplings events on each a control site and illuminated site, respectively. Subsequently we excluded the plant species for which we recorded less than ten interactions during the entire season. For each sampling event, we then quantified the number of interactions each of the 21 plant species had with all pollinators belonging to the three orders mentioned.

The number of interactions between each plant species and pollinator group was analysed using a general linear mixed model (lmer function from the R[43] package lme4[44]) that included plant abundance (scaled logarithm), plant species (21 levels), light treatment (two levels: illuminated site versus control site), pollinator group (three levels: Diptera, Hymenoptera, and Coleoptera), and the interaction between all of them as fixed effects. We further included a random factor pair, which accounted for the paired sampling events (37 levels). To fulfill the model assumptions, the dependent variable was log-transformed.

**Reporting summary.** Further information on research design is available in the Nature Research Reporting Summary linked to this article.

## Data availability
All data analyzed in this study are available in the Zenodo repository at https://zenodo.org/record/4540407#.YCqYPTKg82w[45]. Source data are provided with this paper.

## Code availability
The R code used to analyse the data is available in the Zenodo repository at https://zenodo.org/record/4540407#.YCqYPTKg82w[45].

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

## Acknowledgements
We thank to all who assisted with fieldwork and helped with the identification of insect species. This study was supported by the Swiss National Sciences Foundation, the Federal Institute for the Environment, and the URPP Global Change and Biodiversity of the University of Zürich.

## Author contributions
E.K. and C.F. conceived the study. S.G. and E.K. developed the setup of the field sites and protocols. S.G. and E.K. analyzed the data and wrote the first manuscript draft. All authors reviewed the manuscript.

## Competing interests
The authors declare no competing interests.
