## [Peer Review File · Nature Communications]

Reviewers' Comments:

Reviewer #1:

Remarks to the Author:

I was pleased to be asked to review this manuscript. The authors have previously suggested that the impacts of artificial light at night (ALAN) could permeate to diurnal organisms, so it's great to see some experimental assessment of that hypothesis. In this light, the overall pattern (reductions in frequency of some plant-pollinator interactions at lit sites, increases in others, and no effect in others, all depending on pollinator and plant taxon) is certainly intriguing. I do feel, unfortunately, that the current manuscript suffers a bit from being unable to establish the mechanism behind these effects, though some valid hypotheses are proposed in the discussion and could form a basis for future investigation.

There is, however, a major flaw in the experimental/analytical design which currently erodes my confidence in the results. At face value, it appears (from inspection of Supplementary Table 1) as though a nicely-balanced experiment was designed and conducted in 2016, with 6 dark and 6 lit sites sampled in total. The dataset collected in this sampling period should be robust and I would be pleased to see it analysed. However, sampling also took place separately in 2014 at 8 dark sites (with, to be honest, no real indication in the methodology that this sampling was ever intended to form part of a study of ALAN). No lit sites were sampled in 2014 for comparison (Supp Table 1). Inclusion of data from this sampling period in the analysis creates an enormous problem, because the field methodology is such that plant-pollinator interaction frequency will essentially be a function of pollinator abundance; and pollinator (i.e. insect) abundance can vary hugely from year to year. It is therefore quite possible for effects such as those reported to be generated simply by certain insects (Hymenoptera, some Diptera) being much less abundant in 2016 than 2014, and others (some Diptera) being much more abundant in 2016 than 2014. I cannot see a statistical way around this, and so I strongly recommend the authors exclude this 2014 dataset from their analyses, since it only serves to complicate matters.

An additional concern (which may become irrelevant once reanalysis has taken place!) is that one of the two major effects upon which this manuscript hangs its hat – the effect of lighting on Hymenopteran interactions – is only very marginally significant ($P = 0.047$). Multiple related hypotheses are being tested here (arguably either 20 – one per p-value – or at the least 5 – one per model; Table 1), and if any kind of multiple-testing correction were applied (e.g. Bonferroni or Benjamini-Hochberg correction), this effect would certainly be deemed non-significant. I'm not an advocate of entirely ignoring effects based on Bonferroni-type correction but such correction is probably appropriate here, and the effect should therefore be viewed with an appropriate level of caution. Of course, given I have recommended complete reanalysis in my previous paragraph, this comment may become irrelevant!

Lastly, although I think the manuscript is well-written in general, there are certainly ways in which its readability and comprehension could be improved. First, I would like to see a little more methodological detail introduced in brief into the main text. The nature of a Nat Comms paper is that the methods are read last (if at all) and so basic elements of methodology introduced at appropriate moments do help a lot. For example, could the authors describe very briefly how interactions were sampled at roughly line 41, saying "we recorded 3718 interactions along 100m transects where diurnal [pollinators] actively interacted with receptive flowers of 22 [plants]", instead of the current "we assessed 3718 interactions [between pollinators and plants]". Second, I think the discussion is currently a little skin-deep. At the moment the only thing discussed in detail is the possible mechanisms behind the reported effect, and nothing else. This discussion is valuable but I would like to see the authors also comment on how their results fit into the wider literature on ALAN, and even more broadly, on global change and pollination.

Specific comments

L14: I agree that artificial light at night is rapidly spreading but not that it is relatively new. ALAN has been around for as long as humans have been making fire, so one could argue that it is the oldest of all anthropogenic drivers of change!

L34: Macgregor et al. (2019, *Ecosphere*, doi: 10.1002.ecs2.2550) showed a positive effect of ALAN for plant reproductive output in *Silene latifolia*, in contrast with the Knop et al. study. This contrast seems pertinent in light of your results here (that the positive/negative direction of ALAN effects may vary between different plant species for some pollinator groups).

L44-46: This may need some clarification. At the moment this implies you did two analyses, first including all pollinators and second excluding anything outside the four listed orders. However of course, you actually did five analyses, with a separate analysis for each of the listed orders. I suggest rephrasing to "We did so for plant-pollinator interactions including all pollinators combined, and separately for each of the orders Hymenoptera, Diptera, Coleoptera and Lepidoptera".

L111-113: This statement is slightly confusing here, because the explanation of how plant abundance was quantified is in a later section.

L147-152: Please specify the error distribution used in the models. Also please describe how model fit was assessed and how the most appropriate error distribution was determined.

L160-161: Correct to "right column".

Figures: it is not clear to me why Fig 1 shows mean +/- standard error but Fig 2 shows mean +/- standard deviation – wouldn't it make sense to remain consistent across figures?

Fig 1: It's really unclear to me what this figure shows. The caption doesn't help much as it's not very clearly written. Could this figure be replaced with something closer to the raw data; either model-estimated number of interactions +/- error in each treatment, or effect sizes (per Fig 2)? I would prefer to see some representation of the underlying data. And I would prefer to see all five models (combined and all four orders) presented, rather than just the significant Hymenoptera model.

Fig 2: How is it possible for three species to have a baseline of -0.1 interactions in dark conditions? This possibly suggests that an inappropriate model structure has been used... but currently the model error distribution is not specified in the methods (per my comment above). At a guess, I would say it's Gaussian with sqrt transformation which is not really best practice for this type of data – for count data I'd prefer to see Poisson- or negative binomial-type distributions (i.e. or a related quasi- and/or zero-inflated alternative, as appropriate) since these are all zero-bounded distributions (you can't observe fewer than zero interactions).

Data availability: The current statement is that data is available from the authors upon request. I realise that this is permissible for this type of dataset in Nature Communications so, technically speaking, no change is required. It is always a little disappointing to see it in a journal that promotes Open Science, however. Datasets that are "available upon request" are not really Open, because they depend on (i) the authors remaining willing to make them available in perpetuity; (ii) the authors managing to store the datasets in perpetuity, never mislaying or accidentally deleting any; and (iii) the authors remaining in academia and/or readily contactable in perpetuity (sadly, a biological impossibility!). This is illustrated at present time by the widely-known affair of a certain ecologist, who has been reported to be withholding datasets that were supposedly "available upon request", presumably in order to prevent their reanalysis upon emergence of suspicious patterns in their other papers. Could the authors consider whether they really have a good reason not to make the dataset available in an online repository (e.g. Dryad, Zenodo) upon acceptance?

Reviewer #2:

Remarks to the Author:

This is an interesting study. However, its novelty does not seem to lie principally in the question that it is foremost stated to be focusing on. The stated motivation (lines 36-37) for the work is that 'we still miss evidence that effects of artificial light at night can actually spread to diurnal communities'. But

previous studies, including some that are cited (e.g. ref. 5), have already shown that such spread does occur (e.g. Davies et al. 2017 *Global Change Biol.*). They have found that communities sampled during the day under night-time artificial lighting treatments are different from those under control treatments, including in species composition and abundance structure.

Of course, as has frequently been observed, artificial night-time lighting has a multitude of effects on diurnal species (including on their behaviour, physiology, abundance etc) as well as nocturnal ones, so effects on daytime community structure will almost certainly be widespread.

What is most novel about the present study is its focus on species interactions. One might argue that if night-time artificial lighting is known to influence community structure during the day then it almost inevitably follows that it will also influence these interactions. However, it is nice to see direct confirmation of this.

This said, the present study is using a much more intense light treatment (75 lux immediately under the source) in the experiment that it reports than has been used by the vast majority of experimental studies of its ecological impacts. This is, for example, about ten times brighter than the intensities used in the well-known long-running Dutch study of such impacts. It thus seems designed to maximise the likely impacts on species interactions, rather than – as most others have done – emulate the artificial light conditions that will frequently be experienced by organisms in the vicinity of street lamps.

The study would have been much stronger if it provided some insight into the mechanisms that give rise to alterations in plant-pollinator interactions during the daytime. However, unfortunately, it can only speculate in this regard.

In sum, whilst the work reported in this study is valuable, it would not seem to attain the levels of novelty and insight that one would associate with this journal, and the significance of the findings reported may be doubtful because of the experimental design.

Below we address all of the reviewers' comments and suggestions.

Reviewer #1 (Remarks to the Author):

Thank you very much for your very valuable comments. We believe that aside other improvements, they greatly made our results more robust.

I was pleased to be asked to review this manuscript. The authors have previously suggested that the impacts of artificial light at night (ALAN) could permeate to diurnal organisms, so it's great to see some experimental assessment of that hypothesis. In this light, the overall pattern (reductions in frequency of some plant-pollinator interactions at lit sites, increases in others, and no effect in others, all depending on pollinator and plant taxon) is certainly intriguing. I do feel, unfortunately, that the current manuscript suffers a bit from being unable to establish the mechanism behind these effects, though some valid hypotheses are proposed in the discussion and could form a basis for future investigation.

Thank you for this positive answer. We fully agree that it would be very nice and important to already know the mechanisms behind the found effects. As outlined in the discussion, there are probably different mechanisms causing the indirect effect of artificial light at night on diurnal plant-pollinator interactions. A challenging and important next step is to disentangle them. That is why we believe a prompt publication is important and will stimulate research not only of scientists working in the field of the ecological and evolutionary effects of artificial light at night, but also the work of pollination ecologists, especially the ones that work on the global decline of pollinators and the ecosystem service they provide.

There is, however, a major flaw in the experimental/analytical design which currently erodes my confidence in the results. At face value, it appears (from inspection of Supplementary Table 1) as though a nicely-balanced experiment was designed and conducted in 2016, with 6 dark and 6 lit sites sampled in total. The dataset collected in this sampling period should be robust and I would be pleased to see it analysed. However, sampling also took place separately in 2014 at 8 dark sites (with, to be honest, no real indication in the methodology that this sampling was ever intended to form part of a study of ALAN). No lit sites were sampled in 2014 for comparison (Supp Table 1). Inclusion of data from this sampling period in the analysis creates an enormous problem, because the field methodology is such that plant-pollinator interaction frequency will essentially be a function of pollinator abundance; and pollinator (i.e. insect) abundance can vary hugely from year to year. It is therefore quite possible for effects such as those reported to be generated simply by certain insects (Hymenoptera, some Diptera) being much less abundant in 2016 than 2014, and others (some Diptera) being much more abundant in 2016 than 2014.

I cannot see a statistical way around this, and so I strongly recommend the authors exclude this 2014 dataset from their analyses, since it only serves to complicate matters.

We agree that, as a matter of fact, annual variation in insect abundances can potentially bias our results by confounding the effect of our experimental treatment. We do not really know why we did not think of that – the only thing we had in mind was to make the results more robust by including also the data from another year, thereby forgetting about the aspect raised by reviewer 1. We have now removed the data from 2014 and performed new analyses in which we only included the data from 2016. This resulted in even a stronger effect of artificial light at night on diurnal plant-pollinator interactions.

In addition to the point mentioned by reviewer 1, removing 2014 strengthened our dataset in another way: in 2016 the data was collected using a paired approach, for which we could not statistically account when including also the data from 2014. The paired approach was such that the sampling of a paired control and treatment site (see methods, line 161-163) was done on the same day and at the same time (we did not do this in 2014). We did so in order to account for the fact that diurnal pollinators react very fast to changes of abiotic conditions, i.e., we wanted to minimize the influence of factors other than our treatment on the number of plant-pollinator interactions. As a consequence, we now include a random factor “pair” in our model (see methods, line 192-197). Furthermore, we slightly modified the selection of the plant species which we included in the analyses: in addition to only analyzing plants that were present in at least a minimal number of times on dark sites and illuminated sites (we increased the threshold from four to five, but both give similar results), we added the criteria that the plant received a minimum of 10 visits over the entire season. The latter criteria should exclude those plant species that were very sporadically visited and for which our method of quantifying the interactions may be suboptimal (see methods, line 186-191). Thus, we believe that this new selection is more robust (more replicates per treatment and plant species) and meaningful (a minimal number of pollination visits).

An additional concern (which may become irrelevant once reanalysis has taken place!) is that one of the two major effects upon which this manuscript hangs its hat – the effect of lighting on Hymenopteran interactions – is only very marginally significant ($P = 0.047$). Multiple related hypotheses are being tested here (arguably either 20 – one per p -value – or at the least 5 – one per model; Table 1), and if any kind of multiple-testing correction were applied (e.g. Bonferroni or Benjamini-Hochberg correction), this effect would certainly be deemed non-significant. I'm not an advocate of entirely ignoring effects based on Bonferroni-type correction but such correction is probably appropriate here, and the effect should therefore be viewed with an appropriate level of caution. Of course, given I have recommended complete reanalysis in my previous paragraph, this comment may become irrelevant!

We agree that it is not so nice to have an overall model run with all data included and to subsequently run several models with different subsets of the data. As an elegant solution for the mentioned problem we added an insect group factor, consisting of three levels, namely Diptera, Hymenoptera and Coleoptera (excluding representatives of the Carabidae family). Data of Lepidoptera was excluded as we recorded a total of only 22 interactions. We now thus only present the outcome of one big model.

Lastly, although I think the manuscript is well-written in general, there are certainly ways in which its readability and comprehension could be improved. First, I would like to see a little more methodological detail introduced in brief into the main text. The nature of a Nat Comms paper is that the methods are read last (if at all) and so basic elements of methodology introduced at appropriate moments do help a lot. For example, could the authors describe very briefly how interactions were sampled at roughly line 41, saying “we recorded 3718 interactions along 100m transects where diurnal [pollinators] actively interacted with receptive flowers of 22 [plants]”, instead of the current “we assessed 3718 interactions [between pollinators and plants]”.

We completed the main text to provide more methodological details in the paragraph before the section of the results (line 36-47):

“In 2016, we sampled diurnal plant-pollinator interactions on 12 unmanaged meadows, of which 6 were experimentally illuminated at night using commercial LED street lamps (lit treatment sites) and 6 were left untreated (dark control sites) using a paired approach (see Methods). The paired approach was such that during daytime, we simultaneously quantified plant-pollinator interactions along 100 m transects on a pair of treatment and control sites (37 sampling days with each a control and treatment site sampled simultaneously; see Methods). This allowed minimizing the influence of factors other than the light treatment, which potentially could have influenced the number of plant-pollinator interactions such as weather conditions. In the analysis, we focused on the interactions between 21 naturally occurring plant species regularly present across our study sites, and on insect groups acknowledged to be pollinators, namely Diptera, Hymenoptera and Coleoptera (2384 interactions in total; see Methods). Interactions involving other insect groups or plant species present on only a few sites or in very low abundances were also recorded but not included because their number was too low to be analysed (see Methods).”

Second, I think the discussion is currently a little skin-deep. At the moment the only thing discussed in detail is the possible mechanisms behind the reported effect, and nothing else. This discussion is valuable but I would like to see the authors also comment on how their results fit into the wider literature on ALAN, and even more broadly, on global change and pollination.

We thoroughly revised the discussion and added two paragraphs on how the results fit into the wider literature on ALAN, and on global change and pollination. The two paragraphs are (line 121-138):

“While previous studies brought evidence that artificial light at night disrupts plant-pollinator interactions during the night^{19,20} with potential consequences for the plant-pollinator community during daytime¹⁹, here we bring evidence that such knock-on effects actually occur. In contrast to the few previous studies which focused on such a spreading effect of artificial light at night to the day, and which found either for single species a prolongation of their activity during daytime³⁹ or a shift in the abundance of invertebrate specimens within taxonomic groups⁵, our result expand these findings to species interactions at the community level. Although the effect of artificial light at night varied among plant species and pollinator groups, the alteration of the interactions could have consequences for community stability and functioning.”

and

“Artificial light at night is increasingly recognized as a global change driver⁴⁰. Our results show that further than directly affecting the functioning of night-time ecological activities, the impact is amplified by secondary responses propagating through the system of interacting species thereby affecting other species of the community which are not directly exposed to the global change driver. This is similar to what has been observed with other global change drivers⁴¹ and stresses the need to further investigate the consequences at community level, including the full day and night periods. While the mechanisms underlying this effect of artificial light at night on diurnal plant-pollinator communities still need to be elucidated in future, another research avenue is to look for the interactive effects between artificial light at night and other recognized global change drivers⁴¹.”

Specific comments

L14: I agree that artificial light at night is rapidly spreading but not that it is relatively new. ALAN has been around for as long as humans have been making fire, so one could argue that it is the oldest of all anthropogenic drivers of change!

We agree and removed “new” and specified that it is the drastic spread of this global change driver that happened recently (line 4-5):

“Artificial light at night has rapidly spread around the globe over the last decades, which resulted in a loss of the nightscape in many parts of the world.”

L34: Macgregor et al. (2019, *Ecosphere*, doi: 10.1002.ecs2.2550) showed a positive effect of ALAN for plant reproductive output in *Silene latifolia*, in contrast with the Knop et al. study. This contrast seems pertinent in light of your results here (that the positive/negative direction of ALAN effects may vary between different plant species for some pollinator groups).

We agree with reviewer 1 that it is important to also mention the findings of Macgregor et al (2019) here. We thus now describe the two seemingly contrasting findings as follows (line 26-31):

“Further, artificial light at night has been shown to disrupt nocturnal plant-pollinator interactions¹⁹⁻²⁰, with negative consequences for plant reproductive output¹⁹. Interestingly, also a positive effect of artificial light at night on the reproductive output has recently been reported²¹, suggesting more complex indirect effects of artificial light at night, probably involving diurnal pollinator communities or other organisms such as herbivores or predators.”

L111-113: This statement is slightly confusing here, because the explanation of how plant abundance was quantified is in a later section.

We agree, thank you for pointing this out. We moved the sentence “Most abundant and widespread plant was *Cirsium oleraceum* (Asteraceae), followed by other species being abundant but not present on all sampling sites: *Angelica sylvestris* (Apiaceae), *Eupatorium cannabinum* (Asteraceae), *Erigeron annuus s.l.* (Asteraceae) and *Filipendula ulmaria* (Rosaceae)” to the section in the methods (line 181-184).

L160-161: Correct to “right column”.

Thank you for pointing out this. As we re-run all statistical analyses, we have also changed the figures. Rather than having two figures, we put them all in one figure. Also, we decided not to provide the baseline estimate anymore for several reasons: First, we provide the means of the observed data in the supplementary (we believe it is nicer to also provide the observed data, not only the predicted). Second, we also provide the summary output of our model, where one can look up the predicted baseline information, if needed. It looks nicer and clearer to have all the panels next to each other, without being interrupted by numbers.

Figures: it is not clear to me why Fig 1 shows mean +/- standard error but Fig 2 shows mean +/- standard deviation – wouldn't it make sense to remain consistent across figures?

Fig 1: It's really unclear to me what this figure shows. The caption doesn't help much as it's not very clearly written. Could this figure be replaced with something closer to the raw data; either model-estimated number of interactions +/- error in each treatment, or effect sizes (per Fig 2)? I would prefer to see some representation of the underlying data. And I would prefer to see all five models (combined and all four orders) presented, rather than just the significant Hymenoptera model.

Fig 2: How is it possible for three species to have a baseline of -0.1 interactions in dark conditions? This possibly suggests that an inappropriate model structure has been used... but currently the model error distribution is not specified in the methods (per my comment above). At a guess, I would say it's Gaussian with sqrt transformation which is not really best practice for this type of data – for count data I'd prefer to see Poisson- or negative binomial-type distributions (i.e. or a related quasi- and/or zero-inflated alternative, as appropriate) since these are all zero-bounded distributions (you can't observe fewer than zero interactions).

In response to the comment of reviewer 1, we tried to fit generalized linear mixed effect models assuming Poisson (checking for overdispersion) and Negative Binomial error distribution along with general linear mixed effect models assuming Gaussian distribution and transformations of the response variable to fulfill model assumptions. For the reasons outlined below, we decided in the end to analyze our data with the latter, though we agree with the reviewer, that theoretically a generalized linear mixed effect model would suit count data better.

We did not manage to properly run generalized linear mixed effect models (Poisson error distribution or Negative Binomial error distribution), probably due to having extreme high counts for the plant-honey bee interactions, and many very low counts of interactions with other species. When running generalized linear mixed effect models, we got warnings about the fact that variance-covariance matrix (computed from finite-difference Hessian) is not positive definite or contains NA values. Alternatively the model was not identifiable due to large eigenvalue ratio. Moreover, while testing for overdispersion, we did not get any output at all because PIRLS (penalized iteratively reweighted least squares) step-halving failed to reduce deviance. As a consequence, we got untrustworthy models in which estimates had extremely high standard errors.

General linear mixed effect models assuming Gaussian error distribution, on the other hand, consistently run well. The model in which we square root transformed the data gave the similar effects as when we log-transformed our dependent variable. However, the assumption of normality of the model residuals was nicer met (QQ-plot) with a log transformation than with a square root transformation and we thus decided to analyze our data with a log-transformed response variable.

Response variable square-root transformed

Normal Q-Q Plot

Response variable log-transformed

Normal Q-Q Plot

Data availability: The current statement is that data is available from the authors upon request. I realise that this is permissible for this type of dataset in Nature Communications so, technically speaking, no change is required. It is always a little disappointing to see it in a journal that promotes Open Science, however. Datasets that are “available upon request” are not really Open, because they depend on (i) the authors remaining willing to make them available in perpetuity; (ii) the authors managing to store the datasets in perpetuity, never mislaying or accidentally deleting any; and (iii) the authors remaining in academia and/or readily contactable in perpetuity (sadly, a biological impossibility!). This is illustrated at present time by the widely-known affair of a certain ecologist, who has been reported to be withholding datasets that were supposedly “available upon request”, presumably in order to prevent their reanalysis upon emergence of suspicious patterns in their other papers. Could the authors consider whether they really have a good reason not to make the dataset available in an online repository (e.g. Dryad, Zenodo) upon acceptance?

We will load the dataset to zenodo upon publication.

Reviewer #2 (Remarks to the Author):

Thank you for the points you raised, from which the clarity of our manuscript will definitely benefit.

This is an interesting study. However, its novelty does not seem to lie principally in the question that it is foremost stated to be focusing on. The stated motivation (lines 36-37) for the work is that ‘we still miss evidence that effects of artificial light at night can actually spread to diurnal communities’. But previous studies, including some that are cited (e.g. ref. 5), have already shown that such spread does occur (e.g. Davies et al. 2017 Global Change Biol.). They have found that communities sampled during the day under night-time artificial lighting treatments are different from those under control treatments, including in species composition and abundance structure.

We agree with reviewer 2 that that sentence in lines 36-37 is misleading. We modified it specifying that evidence is missing for species interactions within an entire plant-pollinator community (line 33-34). This is a clear contrast to what has been found previously and potentially has far reaching consequences, especially since it concerns pollinators which provide a very important ecosystem function and services. So we modified the sentence to:

“However, we still miss evidence that effects of artificial light at night can actually spread to diurnal interactions.”

Of course, as has frequently been observed, artificial night-time lighting has a multitude of effects on diurnal species (including on their behaviour, physiology, abundance etc) as well as nocturnal ones, so effects on daytime community structure will almost certainly be widespread. What is most novel about the present study is its focus on species interactions. One might argue that if night-time artificial lighting is known to influence community structure during the day then it almost inevitably follows that it will also influence these interactions. However, it is nice to see direct confirmation of this.

Thank you for this positive comment. We agree, it is novel that we focus on species interactions of an entire plant-pollinator community.

This said, the present study is using a much more intense light treatment (75 lux immediately under the source) in the experiment that it reports than has been used by the vast majority of experimental studies of its ecological impacts. This is, for example, about ten times brighter than the intensities used in the well-known long-running Dutch study of such impacts. It thus seems designed to maximise the likely impacts on species interactions, rather

than – as most others have done – emulate the artificial light conditions that will frequently be experienced by organisms in the vicinity of street lamps.

We respectfully disagree with reviewer 2 about the effectiveness of our installations to emulate real-life situations of light polluted areas. On the contrary, our study was designed to accurately emulate a real situation of disturbance from a common and modern streetlamp model. We selected one of the most widely used commercial LED streetlamps and mounted it on a 6 meters high pole in order to reproduce the lighting conditions of a road crossing. Therefore, the light intensity measured immediately under the source is comparable with the one measured under any street lamp of this kind. In the text we also specified that the value of about 75 lux was measured at less than 2 meters of distance from the lamp, implying that only a very small surface of ground is exposed to such a high light intensity. From the text, methods section: "Light intensity on illuminated sites followed a negative exponential curve as function of the distance from the lamp dropping from 75.73 ± 1.54 lux just under the lamp (<2 m) to 2.67 ± 0.19 lux 10 ± 1.0 m away. Within 20 m light intensity never dropped below 0.045 lux". Since we sampled along a 100 meters long transect, our system was exposed to various light intensities and only few meters of the transect, if ever, was exposed to such high light intensity levels.

The study would have been much stronger if it provided some insight into the mechanisms that give rise to alterations in plant-pollinator interactions during the daytime. However, unfortunately, it can only speculate in this regard.

Demonstrating the propagation of ALAN effects to diurnal species interactions and potentially on ecosystem functions relying on them is the first step for understanding the mechanisms behind the observed pattern. We believe that the most important contribution of this work is that it shows that ALAN effects during the day vary according to the considered plant species and pollinator groups. Also, our results provide a set of expectations on which future studies dealing with plant species used here (common in this type of habitat and therefore good models) might want to rely on. Thus, we consider the present work as the first, important milestone for a new research line that would involve scientists from different fields of study.

In sum, whilst the work reported in this study is valuable, it would not seem to attain the levels of novelty and insight that one would associate with this journal, and the significance of the findings reported may be doubtful because of the experimental design.

Reviewers' Comments:

Reviewer #1:

Remarks to the Author:

I thank the authors for engaging so fully with my previous recommendations. The manuscript is vastly improved, and I'm pleased for the authors that such a major re-analysis has not made their significant effects disappear! The authors have argued the case for their new statistical approach (Gaussian with log-transformation) very clearly; I am generally happier with this approach than sqrt-transformation when count data are involved, but in any case they have clearly been diligent in selecting the most parsimonious model from a number of appropriate options (and I agree that the Q-Q plot for log-transformation is nicer than that for sqrt-transformation).

My assessment is that the revised manuscript remains interesting, and now is also technically sound; it is certainly publishable in its current form. Both myself and reviewer 2 agreed that the lack of a mechanism underpinning the observed effects was a weakness, and I feel it remains an insurmountable weakness, but I fully agree with the authors that publication of the manuscript in its current form may serve to stimulate future work to uncover said mechanism.

My only specific comment refers to the legend for the new Figure 1: it would be beneficial to the readers to include an explanation of exactly what the effect sizes in question mean. Given the model structure (Gaussian/log), I would imagine they mean something like: "interaction frequency at lit sites = interaction frequency at unlit sites $\times e^{\{\text{effect size}\}}$ " (e.g. 1 unlit interaction with an ES of 1 would translate to 2.7 lit interactions).

Reviewer #2:

Remarks to the Author:

The revised version of this manuscript has not really addressed my main concerns with the previous version.

The response to the comments made helps clarify the experimental design and that the lighting treatments may be much more reasonable than seemed on my reading originally to be the case. However, it is still rather unclear just what light treatments the vegetation (and pollinators) was actually experiencing. Attention is drawn to light emissions at 2m from the streetlamps and its decline with greater distances. But is that 2m really relevant in the context of meadow vegetation and pollinators? Surely what is important, and what would enable comparison with other studies (the cause of my original querying 75 lux), is what the organisms were broadly experiencing.

This aside, much more fundamentally, this manuscript remains rather 'thin'. The main finding reported, that artificial nighttime light can have impacts on daytime plant-pollinator interactions is a useful advance. But it is not a huge one, given that, as the authors now explicitly recognise, changes in daytime invertebrate species composition and abundances have previously been found to result from artificial nighttime light. Moreover, the effects on plant-pollinator interactions that are found are not consistent, with differences between plant species and pollinator groups that are difficult to generalise from this particular study.

As both reviewers observed with regard to the first version of the manuscript, what is really needed at this stage is some insight into the mechanisms that result in observed changes in plant-pollinator interactions by artificial nighttime lighting, and in the variation in responses. This is entirely lacking. Indeed, the major part of the text of the manuscript is now speculation as to what those mechanisms might be rather than discussion of the results per se. In the response to reviewers the authors argue that their manuscript should be published in order to stimulate the required mechanistic studies. But

one does expect that papers published in Nature Communications contribute substantially more than principally a stimulus for further work.

Below we address all of the reviewers' comments and suggestions.

Reviewer #1 (Remarks to the Author):

I thank the authors for engaging so fully with my previous recommendations. The manuscript is vastly improved, and I'm pleased for the authors that such a major re-analysis has not made their significant effects disappear! The authors have argued the case for their new statistical approach (Gaussian with log-transformation) very clearly; I am generally happier with this approach than sqrt-transformation when count data are involved, but in any case they have clearly been diligent in selecting the most parsimonious model from a number of appropriate options (and I agree that the Q-Q plot for log-transformation is nicer than that for sqrt-transformation).

My assessment is that the revised manuscript remains interesting, and now is also technically sound; it is certainly publishable in its current form. Both myself and reviewer 2 agreed that the lack of a mechanism underpinning the observed effects was a weakness, and I feel it remains an insurmountable weakness, but I fully agree with the authors that publication of the manuscript in its current form may serve to stimulate future work to uncover said mechanism.

My only specific comment refers to the legend for the new Figure 1: it would be beneficial to the readers to include an explanation of exactly what the effect sizes in question mean. Given the model structure (Gaussian/log), I would imagine they mean something like: "interaction frequency at lit sites = interaction frequency at unlit sites $\times e^{\text{effect size}}$ " (e.g. 1 unlit interaction with an ES of 1 would translate to 2.7 lit interactions).

Thank you for your positive feedback and the very valuable suggestion regarding legend. We have added the following explanation to the legend of Fig.1:

As the response variable was log-transformed to fulfill the model assumptions (Gaussian error distribution), the back transformed number of interactions at lit sites equals to the number of interactions at unlit sites $\times \exp(\text{effect size})$. For example, an average of 10 interactions at unlit sites with an effect size of -1 translates in $10 \times \exp(-1)$ interactions at lit sites (i.e., 3.68 interactions).

Reviewer #2 (Remarks to the Author):

The revised version of this manuscript has not really addressed my main concerns with the previous version.

The response to the comments made helps clarify the experimental design and that the lighting treatments may be much more reasonable than seemed on my reading originally to be the case. However, it is still rather unclear just what light treatments the vegetation (and pollinators) was actually experiencing. Attention is drawn to light emissions at 2m from the streetlamps and its decline with greater distances. But is that 2m really relevant in the context of meadow vegetation and pollinators? Surely what is important, and what would enable comparison with other studies (the cause of my original querying 75 lux), is what the organisms were broadly experiencing.

Thank you for addressing this issue again. We apologize for not having understood your main criticism in the first place. As the pollinator sampling was performed along transects around the streetlamp, the light intensity varied as we got further away from the streetlamp and we agree that providing light intensity 2m away from the street lamp was not the most meaningful. After having discussed it once more, we decided to add a figure to the Supplementary Information (Supplementary Fig. 2), which

shows the light intensity gradient as a function of distance from the street lamp, and we discuss it in the main text as follows:

Light intensity on illuminated sites followed a negative exponential curve as function of the distance from the lamp dropping from 75.73 ± 1.54 lux just under the lamp (<2 m) to 2.67 ± 0.19 lux 10±1.0 m away (Supplementary Fig. 2). Transect for pollinator sampling were 100 m long with the street lamp in the middle, resulting in pollinator sampling all along the light intensity gradient.

We hope that with this we are able to address your main concern.

This aside, much more fundamentally, this manuscript remains rather 'thin'. The main finding reported, that artificial nighttime light can have impacts on daytime plant-pollinator interactions is a useful advance. But it is not a huge one, given that, as the authors now explicitly recognise, changes in daytime invertebrate species composition and abundances have previously been found to result from artificial nighttime light. Moreover, the effects on plant-pollinator interactions that are found are not consistent, with differences between plant species and pollinator groups that are difficult to generalise from this particular study.

As both reviewers observed with regard to the first version of the manuscript, what is really needed at this stage is some insight into the mechanisms that result in observed changes in plant-pollinator interactions by artificial nighttime lighting, and in the variation in responses. This is entirely lacking. Indeed, the major part of the text of the manuscript is now speculation as to what those mechanisms might be rather than discussion of the results per se. In the response to reviewers the authors argue that their manuscript should be published in order to stimulate the required mechanistic studies. But one does expect that papers published in Nature Communications contribute substantially more than principally a stimulus for further work.

Thank you for recognizing that it is an important finding and that more work is needed on the mechanisms causing the inconsistent pattern. We agree that it would be simpler if the pattern would go clearly in one direction. At the same time, we are not surprised that it is how it is as the mechanisms driving the pattern are certainly diverse and species-specific, and hence the direction of the outcome most likely varies depending on the plant and insect species involved.